# HPO-B: A Large-Scale Reproducible Benchmark for Black-Box HPO based on OpenML

**Sebastian Pineda Arango**[*]
University of Freiburg
pineda@cs.uni-freiburg.de

**Hadi S. Jomaa**[*]
University of Hildesheim
hsjomaa@ismll.uni-hildesheim.de

**Martin Wistuba**[†]
Amazon Research
marwistu@amazon.com

**Josif Grabocka**
University of Freiburg
grabocka@cs.uni-freiburg.de

## Abstract

Hyperparameter optimization (HPO) is a core problem for the machine learning community and remains largely unsolved due to the significant computational resources required to evaluate hyperparameter configurations. As a result, a series of recent related works have focused on the direction of transfer learning for quickly fine-tuning hyperparameters on a dataset. Unfortunately, the community does not have a common large-scale benchmark for comparing HPO algorithms. Instead, the *de facto* practice consists of empirical protocols on arbitrary small-scale meta-datasets that vary inconsistently across publications, making reproducibility a challenge. To resolve this major bottleneck and enable a fair and fast comparison of black-box HPO methods on a level playing field, we propose **HPO-B**, a new large-scale benchmark in the form of a collection of meta-datasets. Our benchmark is assembled and preprocessed from the OpenML repository and consists of 176 search spaces (algorithms) evaluated sparsely on 196 datasets with a total of 6.4 million hyperparameter evaluations. For ensuring reproducibility on our benchmark, we detail explicit experimental protocols, splits, and evaluation measures for comparing methods for both non-transfer, as well as, transfer learning HPO.

## 1 Introduction

Hyperparameter Optimization (HPO) is arguably the major open challenge for the machine learning community due to the expensive computational resources demanded to evaluate configurations. As a result, HPO and its broader umbrella research area, AutoML, have drawn particular interest over the past decade [2, 18, 30, 31]. Black-box HPO is a specific sub-problem that focuses on the case where the function to be optimized (e.g. the generalization performance of an algorithm) is unknown, non-differentiable with respect to the hyperparameters, and intermediate evaluation proxies are not computable (opposed to gray-box HPO [23] which accesses intermediate performance measurements).

Although black-box HPO is a core problem, existing solutions based on parametric surrogate models for estimating the performance of a configuration overfit the limited number of evaluated configurations. As a result, the AutoML community has recently invested efforts in resolving the sample-inefficiency of parametric surrogates via meta- and transfer-learning [13, 19, 27, 28, 33, 35, 38].

---

[*]Equal contribution
[†]Work done prior joining Amazon Research

35th Conference on Neural Information Processing Systems (NeurIPS 2021) Track on Datasets and Benchmarks.

Unfortunately, despite the promising potential of transfer-learning in black-box HPO, the impact of such algorithms is hindered by their poor experimental reproducibility. Our personal prior research experience, as well as the feedback from the community, highlight that reproducing and generalizing the results of transfer-learning HPO methods is challenging. In essence, the problem arises when the results of a well-performing method in the experimental protocol of a publication either can not be replicated; or when the method underperforms in a slightly different empirical protocol. We believe that a way of resolving this negative *impasse* is to propose a new public large-scale benchmark for comparing HPO methods, where the exact training/validation/test splits of the meta-datasets, the exact evaluation protocol, and the performance measures are well-specified. The strategy of adopting benchmarks is a trend in related areas, such as in computer vision [8], or NAS [9, 40].

In this perspective, we present **HPO-B**[3], the largest public benchmark of meta-datasets for black-box HPO containing 6.4M hyperparameter evaluations across 176 search spaces (algorithms) and on 196 datasets in total. The collection is derived from the raw data of OpenML [32], but underwent an extensive process of cleaning, preprocessing and organization (Section 5). Additionally, we offer off-the-shelf ready variants of the benchmark that are adapted for both non-transfer, as well as transfer HPO experiments, together with the respective evaluation protocols (Section 6). This large, diverse, yet plug-and-play benchmark can significantly boost future research in black-box HPO.

## 2  Terminology

To help the reader navigate through our paper, we present the compact thesaurus of Table 1 for defining the vernacular of the HPO community.

| Term | Definition |
|---|---|
| Configuration | Specific settings/values of hyperparameters |
| Search space | The domain of a configuration: scale and range of each hyperparameter's values |
| Response | The performance of an algorithm given a configuration and dataset |
| Surrogate | A (typically parametric) function that approximates the response |
| Seed | Set of initial configurations used to fit the initial surrogate model |
| Black-box | The response is an unknown and non-differentiable function of a configuration |
| Task | An HPO problem given a search space and a dataset |
| Evaluation | The measured response of a configuration on a dataset |
| Trial | An evaluation on a task during the HPO procedure |
| Meta-dataset | Collection of *recorded* evaluations from different tasks on a search space |
| Meta-instance | An evaluation in the meta-dataset for one of the tasks |
| Meta-feature | Descriptive attributes of a dataset |
| Source tasks | In a meta- or transfer-learning setup refers to the *known* tasks we *train from* |
| Target tasks | In a meta- or transfer-learning setup refers to the *new* tasks we *test on* |
| Benchmark | **New definition:** Collection of meta-datasets from different search spaces |

Table 1: A thesaurus of the common HPO terminology used throughout this paper

## 3  Related Work

**Non-transfer black-box HPO**: The mainstream paradigm in HPO relies on surrogates to estimate the performance of hyperparameter configurations. For example, [2] were the first to propose Gaussian Processes (GP) as surrogates. The same authors also propose a Tree Parzen Estimator (TPE) for computing the non-parametric densities of the hyperparameters given the observed performances. Both approaches achieve a considerable lift over random [3] and manual search. To address the cubic run-time complexity of GPs concerning the number of evaluated configurations, DNGO [30] trains neural networks for generating adaptive basis functions of hyperparameters, in combination with a Bayesian linear regressor that models uncertainty. Alternatively, SMAC [18] represents the surrogate as a random forest, and BOHAMIANN [31] employs Bayesian Neural Networks instead of plain neural networks to estimate the uncertainty of a configuration's performance. For an extensive study

---
[3]The benchmark is publicly available at `https://github.com/releaunifreiburg/HPO-B`

on non-transfer Bayesian Optimization techniques for HPO, we refer the readers to [6, 29] that study the impact of the underlying assumptions associated with black-box HPO algorithms.

**Transfer black-box HPO**: To expedite HPO, it is important to leverage information from existing evaluations of configurations from prior tasks. A common approach is to capture the similarity between datasets using meta-features (i.e. descriptive dataset characteristics). Meta-features have been used as a warm-start initialization technique [14, 20], or as part of the surrogate directly [1]. Transfer learning is also explored through the weighted combination of surrogates, such as in TST-R [38], RGPE [13], and TAF-R [39]. Another direction is learning a shared surrogate across tasks. ABLR optimizes a shared hyperparameter embedding with separate Bayesian linear regressors per task [24], while GCP [27] maps the hyperparameter response to a shared distribution with a Gaussian Copula process. Furthermore, FSBO [35] meta-learns a deep-kernel Gaussian Process surrogate, whereas DMFBS incorporates the dataset context through end-to-end meta-feature networks [20].

**Meta-datasets**: The work by Wistuba et al. [37] popularised the usage of meta-dataset benchmarks with pre-computed evaluations for the hyperparameters of SVM (288 configurations) and Adaboost (108 configurations) on 50 datasets; a benchmark that inspired multiple follow-up works [13, 34]. Existing attempts to provide HPO benchmarks deal only with the non-transfer black-box HPO setup [10], or the gray-box HPO setup [12]. As they contain results for one or very few datasets per search space, they cannot be used for the evaluation of transfer black-box HPO methods. Nevertheless, there is a trend in using evaluations of search spaces from the OpenML repository [15], which contains evaluations reported by an open community, as well as large-scale experiments contributed by specific research labs [4, 22]. However, the choice of OpenML search spaces in publications is ad-hoc: one related work uses SVM and XGBoost [24], a second uses GLMNet and SVM [35], while a third paper uses XGBoost, Random Forest and SVM [25]. We assess that the community *(i)* inconsistently cherry-picks (assuming *bona fides*) search spaces, with *(ii)* arbitrary train/validation/test splits of the tasks within the meta-dataset, and *(iii)* inconsistent preprocessing of hyperparameters and responses. In our experiments, we observed that existing methods do not generalize well on new meta-datasets.

**Our Novelty:** As a remedy, we propose a novel benchmark derived from OpenML [15], that resolves the existing reproducibility issues of existing non-transfer and transfer black-box HPO methods, by ensuring a fairly-reproducible empirical protocol. The contributions of our benchmark are multi-fold. First of all, we remove the confounding factors induced by different meta-dataset preprocessing pipelines (e.g. hyperparameter scaling and transformations, missing value imputations, one-hot encodings, etc.). Secondly, we provide a specified collection of search spaces, with specified datasets and evaluations. Furthermore, for transfer learning HPO methods, we also provide pre-defined training/validation/testing splits of tasks. For experiments on the test tasks, we additionally provide 5 seeds (i.e. 5 sets of initial hyperparameters to fit the initial surrogate) with 5 hyperparameter configurations, each. We also highlight recommended empirical measures for comparing HPO methods and assessing their statistical significance in Section 6. In that manner, the results of different papers that use our benchmark can be compared directly without fearing the confounding factors. Table 2 presents a summary of the descriptive statistics of meta-datasets from prior literature. To the best of our awareness, the proposed benchmark is also richer (in the number of search spaces and their dimensionality) and larger (in the number of evaluations) than all the prior protocols.

| Paper | Venue/Year | # Search Spaces | # Datasets | # HPs | # Evals. |
|---|---|---|---|---|---|
| [1] | ICML '13 | 1 | 29 | 2 | 3K |
| [37] | DSAA '15 | 2 | 50 | 2, 4 | 20K |
| [14] | AAAI '15 | 3 | 57 | 4, 5 | 93K |
| [36] | ECML-PKDD '15 | 17 | 59 | 1-7 | 1.3M |
| [24] | NeurIPS '18 | 2 | 30 | 4, 10 | 655K |
| [27] | ICML '20 | 4 | 26 | 6, 9 | 343K |
| [20] | DMKD '21 | 1 | 120 | 7 | 414K |
| [35] | ICLR '21 | 3 | 80 | 2, 4 | 864K |
| Our HPO-B-v1 | - | 176 | 196 | 1-53 | 6.39M |
| Our HPO-B-v2/-v3 | - | 16 | 101 | 2-18 | 6.34M |

Table 2: Summary statistics for various meta-datasets considered in prior works.

# 4    A Brief Explanation of Bayesian Optimization Concepts

As we often refer to HPO methods, in this section we present a brief coverage of Bayesian Optimization as the most popular HPO method for black-box optimization. HPO aims at minimizing the function $f : \mathcal{X} \rightarrow \mathbb{R}$ which maps each hyperparameter configuration $\mathbf{x} \in \mathcal{X}$ to the validation loss obtained when training the machine learning model using $\mathbf{x}$. Bayesian Optimization keeps track of all evaluated hyperparameter configurations in a history $\mathcal{D} = \{(\mathbf{x}_i, y_i)\}_i$, where $y_i \sim \mathcal{N}(f(\mathbf{x}_i), \sigma_n^2)$ is the (noisy) response which can be heteroscedastic [17] in real-world problems [6]. A probabilistic model, the so-called surrogate model, is used to approximate the behavior of the response function. Gaussian Processes are a common choice for the surrogate model [26, 29]. Bayesian Optimization is an iterative process that alternates between updating the surrogate model as described above and selecting the next hyperparameter configuration. The latter is done by finding the configuration which maximizes an acquisition function, which scores each feasible hyperparameter configuration using the surrogate model by finding a trade-off between exploration and exploitation. Arguably, the most popular acquisition function is the Expected Improvement [21]. The efficiency of Bayesian Optimization depends on the surrogate model's ability to approximate the response function. However, this is a challenging task since every optimization starts with no or little knowledge about the response function. To overcome this cold-start problem, transfer methods have been proposed, which leverage information from other tasks of the same search space.

# 5    Benchmark Description

The benchmark HPO-B is a collection of meta-datasets collected from OpenML [15] with a diverse set of search spaces. We present three different versions of the benchmark, as follows:

- **HPO-B-v1:** The raw benchmark of all 176 search spaces;

- **HPO-B-v2:** Subset of 16 search-spaces with the biggest amount of evaluations;

- **HPO-B-v3:** Split of HPO-B-v2 into training, validation and testing.

When assembling the benchmark HPO-B-v1 we noticed that most of the evaluations are reported for a handful of popular search spaces, in particular, we noticed that 9% of the top meta-datasets include 99.3% of the evaluations. As a result, we created a second version HPO-B-v2 that includes only the frequent meta-datasets that have at least 10 datasets with at least 100 evaluations per dataset (Section 5.1). Furthermore, as we clarified in Section 3 a major reproducibility issue of the related work on transfer HPO is the lack of clear training, validation, and test splits for the meta-datasets. To resolve this issue, we additionally created HPO-B-v3 as a derivation of HPO-B-v2 with pre-defined splits of the training, validation, and testing tasks for every meta-dataset, in addition to providing initial configurations (seeds) for the test tasks. The three versions were designed to fulfill concrete purposes with regards to different types of HPO methods. For non-transfer black-box HPO methods, we recommend using HPO-B-v2 which offers a large pool of HPO tasks. Naturally, for transfer HPO tasks we recommend using HPO-B-v3 where meta-datasets are split into training, validation, and testing. We still are releasing the large HPO-B-v1 benchmark to anticipate next-generation methods for heterogeneous transfer learning techniques that meta-learn surrogates across different search spaces, where all 176 meta-datasets might be useful despite most of them having few evaluations.

Concretely, HPO-B-v3 contains the set of filtered search spaces of HPO-B-v2, which are specially split into *four* sets: meta-train, meta-validation. meta-test and an augmented version of the meta-train dataset. Every split contains different datasets from the same search space. We distributed the datasets per search space as 80% of the datasets to meta-train, 10% to meta-validation, and 10% to meta-test, respectively. A special, augmented version of the meta-train is created by adding all other search space evaluations from HPO-B-v1 that are not part of HPO-B-v3. On the other hand, in HPO-B-v3 we also provide seeds for initializing the HPO. They are presented as five different sets of five initial configurations to be used by a particular HPO method. By providing five different seeds we decrease the random effect of the specific initial configurations. To ease the comparison among HPO methods, we suggest using the recommended initial configurations for testing. Although, we admit that some algorithms proposing novel warm-starting strategies might need to bypass the recommended initial configurations.

## 5.1 Benchmark summary

The created benchmark contains 6,394,555 total evaluations across 176 search spaces that are sparsely evaluated on 196 datasets. By accounting for the search spaces that comply with our filtering criteria (at least 10 datasets with 100 evaluations), we obtain HPO-B-v2 with 16 different search spaces and 6,347,916 evaluations on 101 datasets. Notice that the benchmark does not include evaluations for all datasets in every search space. The number of dimensions, datasets, and evaluations per search space is listed in Table 3. An additional description of the rest of all the search spaces in HPO-B-v1 is presented in the Appendix. In addition, Table 3 shows the description of the meta-dataset splits according to the HPO-B-v3.

| Search Space | ID | #HPs | Meta-Train | | Meta-Validation | | Meta-Test | |
|---|---|---|---|---|---|---|---|---|
| | | | #Evals. | #DS | #Evals. | #DS | #Evals. | #DS |
| rpart.preproc (16) | 4796 | 3 | 10694 | 36 | 1198 | 4 | 1200 | 4 |
| svm (6) | 5527 | 8 | 385115 | 51 | 196213 | 6 | 354316 | 6 |
| rpart (29) | 5636 | 6 | 503439 | 54 | 184204 | 7 | 339301 | 6 |
| rpart (31) | 5859 | 6 | 58809 | 56 | 17248 | 7 | 21060 | 6 |
| glmnet (4) | 5860 | 2 | 3100 | 27 | 598 | 3 | 857 | 3 |
| svm (7) | 5891 | 8 | 44091 | 51 | 13008 | 6 | 17293 | 6 |
| xgboost (4) | 5906 | 16 | 2289 | 24 | 584 | 3 | 513 | 2 |
| ranger (9) | 5965 | 10 | 414678 | 60 | 73006 | 7 | 83597 | 7 |
| ranger (5) | 5970 | 2 | 68300 | 55 | 18511 | 7 | 19023 | 6 |
| xgboost (6) | 5971 | 16 | 44401 | 52 | 11492 | 6 | 19637 | 6 |
| glmnet (11) | 6766 | 2 | 599056 | 51 | 210298 | 6 | 310114 | 6 |
| xgboost (9) | 6767 | 18 | 491497 | 52 | 211498 | 7 | 299709 | 6 |
| ranger (13) | 6794 | 10 | 591831 | 52 | 230100 | 6 | 406145 | 6 |
| ranger (15) | 7607 | 9 | 18686 | 58 | 4203 | 7 | 5028 | 7 |
| ranger (16) | 7609 | 9 | 41631 | 59 | 8215 | 7 | 9689 | 7 |
| ranger (7) | 5889 | 6 | 1433 | 20 | 410 | 2 | 598 | 2 |

Table 3: Description of the search spaces in HPO-B-v3; "#HPs" stands for the number of hyperparameters, "#Evals." for the number of evaluations in a search space, while "#DS" for the number of datasets across which the evaluations are collected. The search spaces are named with the respective OpenML version number (in parenthesis), and their original names are preceded by *mlr.classif*.

## 5.2 Preprocessing

The OpenML-Python API [16] was used to download the experiment data from OpenML [15]. We have collected all evaluations (referred to as runs in OpenML) tagged with `Verified_Supervised_Classification` available until April 15, 2021.

While the hyperparameter configuration was directly available for many evaluations, some of them had to be parsed from WEKA arguments (e.g. `weka.filters.unsupervised.attribute.RandomProjection -P 16.0 -R 42 -D Sparse1`). A small percentage (<0.001%) of these were too complex in structure to be automatically parsed, so they were discarded. Duplicate responses for the same hyperparameter configuration have been resolved by keeping only one random response. Finally, all tasks with less than five observations were also discarded.

All categorical hyperparameters were one-hot encoded, taking into account all categories that occur in the different datasets for a search space. Missing values have been replaced with zeros and the corresponding missing indicator (a new feature) has been set to one. Hyperparameters that had the same value for all configurations in a search space were dropped. We manually decided which hyperparameters required log-scaling by inspecting the distributions of each hyperparameter in each space (considerable manual effort). Finally, the hyperparameter ranges were scaled to $[0, 1]$. Further details on the pre-processing are explained in Appendix G.

### 5.3 Benchmark JSON schema

The benchmark is offered as easily accessible JSON files. The first-level key of each JSON schema corresponds to the search space ID, whereas the second-level key specifies the dataset ID. By accessing the JSON schema with the search space $s$ and the dataset $t$, we obtain the meta-dataset $\mathcal{D}^{(s,t)} = \{(\mathbf{x}_i^{(s,t)}, y_i^{(s,t)})\}_i, \mathbf{x}_i^{(s,t)} \in \mathcal{X}^{(s)}$. The meta-dataset exhibits the following structure, where $N$ denotes the number of evaluations available for the specific task:

```
{search_space_ID: {dataset_ID:{X:[[x_1],...,[x_N]], y:[[y_1],...,[y_N]]}}}
```

The initialization seeds are similarly provided as a JSON schema, where the third-level subschema has 5 keys whose values are the indices of the samples to use as initial configurations.

### 5.4 An additional continuous variant of HPO-B

OpenML [15] offers only discrete evaluations of hyperparameter configurations. Continuous HPO search methods are not applicable out-of-the-box on the discrete meta-datasets of HPO-B, because evaluations are not present for every possible configuration in a continuous space. To overcome this limitation, we release an additional continuous version of HPO-B based on task-specific surrogates. For every task, we fit an XGBoost [5] regression model with a maximum depth of 6 and two cross-validated hyper-hyperparameters, concretely the learning rate and the number of rounds. We train the surrogates to approximate the observed response values of the evaluated configurations on each task. As a result, for any arbitrary configuration in the continuous space, the approximate evaluation of a configuration's response is computed through the estimation of the respective task's surrogate. Furthermore, a download link to the trained surrogate models is also provided in the repository [4].

## 6 Recommended Experimental Protocol

One of the primary purposes of HPO-B is to standardize and facilitate the comparison between HPO techniques on *a level playing field*. In this section, we provide two specific recommendations: which benchmark to use for a type of algorithm and what metrics to use for comparing results.

**Evaluation Metrics**   We define the average normalized regret at trial $e$ (a.k.a. average distance to the minimum) as $\min_{x \in \mathcal{X}_e^{(s,t)}} \left( f^{(s,t)}(x) - y_{\min}^* \right) / \left( y_{\max}^* - y_{\min}^* \right)$ with $\mathcal{X}_e^{(s,t)}$ as the set of hyperparameters that have been selected by a HPO method up to trial $e$, with $y_{\min}^*$ and $y_{\max}^*$ as the best and worst responses, respectively. The average rank represents the mean across tasks of the ranks of competing methods computed using the test accuracies of the best configuration until the $e$-th trial. Results across different search spaces are computed by a simple mean over the search-space-specific results.

**Non-Transfer Black-Box HPO**   Methods should be compared on all the tasks in HPO-B-v2 and for each of the five initial configurations. The authors of future papers should report the normalized regret and the mean ranks for all trials from 1 to 100 (excluding the seeds). We recommend that the authors show both aggregated and per search-space (possibly moved to the appendix) mean regret and mean rank curves for trials ranging from 1 to 100. In other words, as many runs as the number of tasks for given space times the number of initialization seeds. To assess the statistical significance of methods, we recommend that critical difference diagrams [7] be computed for the ranks of all runs @25, @50, and @100 trials.

**Transfer Black-Box HPO**   Methods should be compared on the meta-data splits contained in HPO-B-v3. All competing methods should use exactly the evaluations of the provided meta-train datasets for meta- and transfer-learning their method, and tune the hyper-hyperparameters on the evaluations of the provided meta-validation datasets. In the end, the competing methods should be tested on the provided evaluations of the meta-test tasks. As our benchmark does not have pre-computed responses for all possible configurations in a space, the authors either *(i)* need to adapt their HPO acquisitions and suggest the next configuration only from the set of the pre-computed configurations for each

---

[4]`https://github.com/releaunifreiburg/HPO-B`

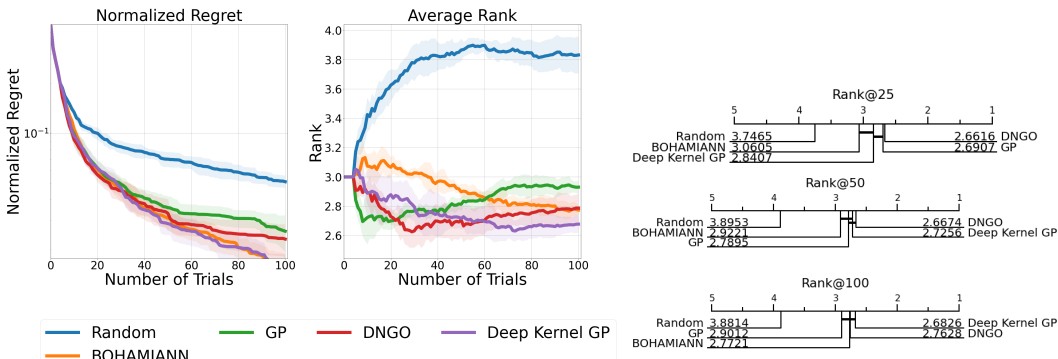

Figure 1: **Aggregated** comparisons of normalized regret and mean ranks across all search spaces for the **non-transfer** HPO methods on HPO-B-v2

specific meta-test task, or *(ii)* use the continuous variant of HPO-B. Additionally, we recommend that the authors present (see details in the paragraph above) regret and rank plots, besides the critical difference diagrams @25, @50, and @100 trials. If a future transfer HPO method proposes a novel strategy for initializing configurations, for the sake of reproducibility we still recommend showing additional results with our initial configurations.

# 7  Experimental Results

The benchmark is intended to serve as a new standard for evaluating non-transfer and transfer black-box HPO methods. In the following, we will compare different methods according to our recommended protocol described in Section 6. This is intended to demonstrate the usefulness of our benchmark, while at the same time serving as an example for the aforementioned recommendations on comparing baselines and presenting results.

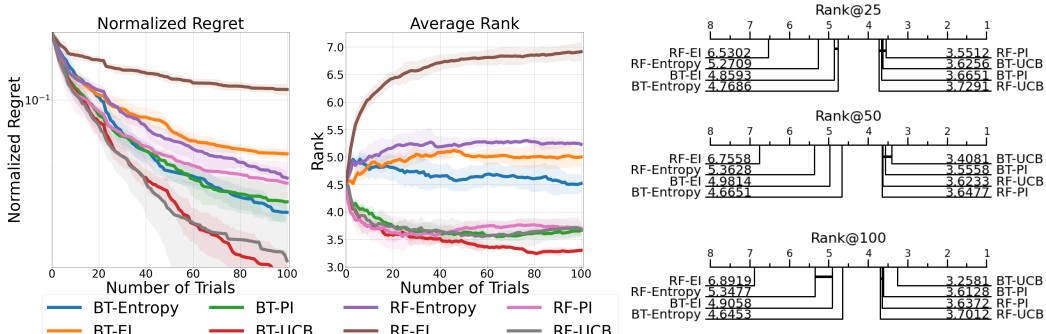

Figure 2: **Aggregated** comparisons of different surrogates and acquisition functions for **non-transfer** HPO tree-based methods on HPO-B-v2; BT stands for Boosted Trees, RF for Random Forests, EI for Expected Improvement, UCB for Upper Confidence Bound, and PI for Probabiliy of Improvement.

## 7.1  Non-transfer Black-Box HPO

First, we compare Random Search, DNGO, BOHAMIANN, Gaussian Process (GP) with Matérn 3/2 kernel, and Deep Gaussian Process (FSBO [35] without pre-training) on HPO-B-v2 in the non-transfer scenario. As recommended by us earlier, in Figure 10 we report aggregated results for normalized regret, average rank, and critical difference plots. In addition, we report in Figure 1 the aggregated normalized regret per search space. The values in the figures for the number of trials equal to 0 correspond to the result after the five initialization steps. According to Figure 1, BOHAMIANN and Deep GP achieve comparable aggregated normalized regret across all search spaces, which suggests that both methods are equally well-suited for the tasks. The average rank and the critical difference

plot paint a different picture, in which Deep GP and DNGO achieve better results. This discrepancy arises because each metric measures different performance aspects on different tasks, so it's important to report both. As can be seen in Figure 11, Deep GP achieves better results than the GP in most of the tasks, which leads to a better average ranking. However, as we can see in Figure 10, the regrets are observed at heterogeneous scales that can skew the overall averages. In some cases where BOHAMIANN outperforms Deep GP (e.g. search spaces 5527, 5859, and 5636), the difference in normalized regret is evident, due to the nature of the search space, whereas in cases where it is the other way around, however, the difference is only slightly less evident (e.g. search spaces 4796, 5906, and 7609). An important aspect of HPO is the choice of the surrogate function and acquisition. Figure 2 presents an ablation of typical combinations and shows the accuracy of the Boosted Tree as a surrogate.

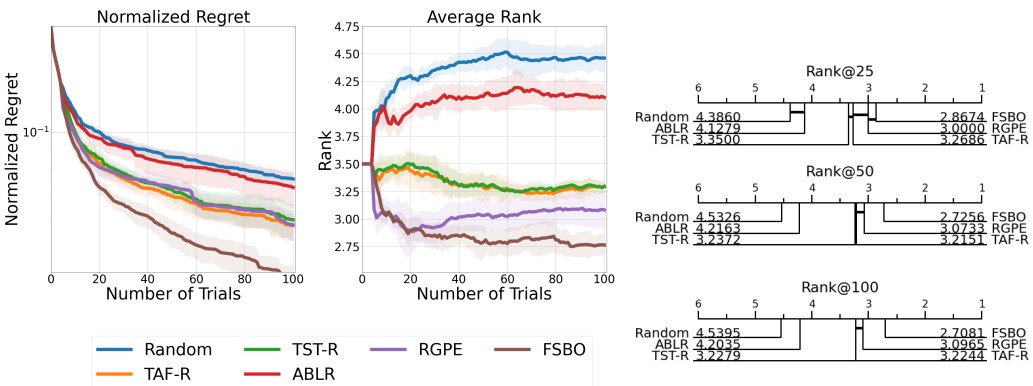

Figure 3: **Aggregated** comparisons of normalized regret and mean ranks across all search spaces for the **transfer learning** HPO methods on HPO-B-v3

## 7.2 Transfer Black-Box HPO

Finally, we compare RGPE [13], ABLR [24], TST-R [38], TAF-R [39], and FSBO [35] on HPO-B-v3 in the transfer scenario. All hyper-hyperparameters were optimized on the meta-validation datasets and we report results aggregated across all test search spaces in terms of normalized regret and average rank in Figure 3. The results per search space for normalized regret and average rank are given in Figure 9 and Figure 12, respectively. FSBO shows improvements over all the compared methods for the normalized regret metric and average rank metric. On the other hand, RGPE is seemingly performing similar to TST-R and TAF-R for the average regret, but performs significantly better for the average rank metric. The explanation is the same as for our last experiment and can mainly be traced back to the strong performance of RGPE in search spaces 5971 and 5906. Such behaviors strengthen our recommendations of Section 6 for showing results in terms of both the ranks and the normalized regrets, as well as the ranks' statistical significance.

## 7.3 Comparing Non-Transfer vs. Transfer Black-Box HPO

We provide a cumulative comparison of both non-transfer and transfer black-box methods in Figure 4, for demonstrating the benefit of transfer learning in HPO-B-v3. We see that the transfer methods (FSBO, RGPE, TST-R, TAF-R) achieve significantly better performances than the non-transfer techniques (GP, DNGO, BOHAMIANN, Deep Kernel GP). On the average rank plot and the associated Critical Difference diagrams, we notice that FSBO [35] achieves significantly better results than all baselines, followed by RGPE [13]. A detailed comparison of the ranks per search-space is presented in the supplementary material. In particular, the direct gain of transfer learning can be observed by the dominance that FSBO has over *Deep Kernel GP*, considering that both use exactly the same surrogate model and the same acquisition function. In comparison, the deep kernel parameters in FSBO are initialized from the solution of a meta-learning optimization conducted on the meta-train tasks of HPO-B-v3 (transfer), while the parameters of *Deep Kernel GP* are initialized randomly (no transfer).

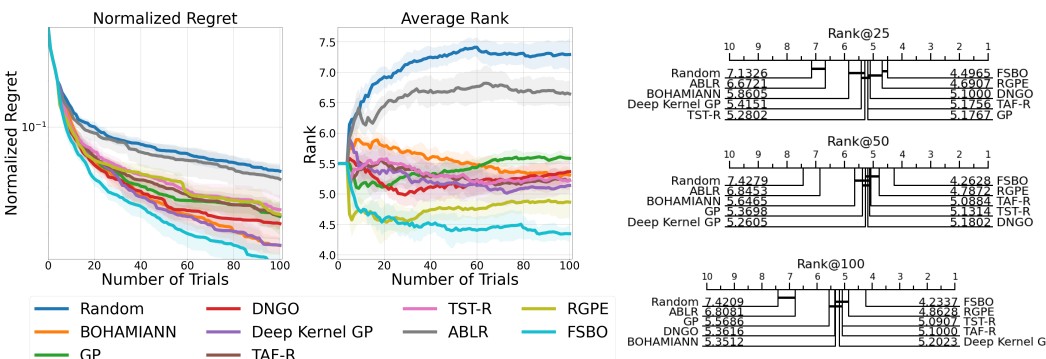

Figure 4: Comparisons of normalized regret and mean ranks across all search spaces for the **transfer learning** and **non-transfer** HPO methods on HPO-B-v3

## 7.4 Validating the Continuous HPO-B Benchmark

We further show that the surrogate-based continuous variant of HPO-B (Section 5.4) provides a benchmark where HPO methods achieve similar performances compared to the discrete HPO-B. We present the results of three typical non-transfer HPO methods (Gaussian Process (GP), Deep Kernel GP, and Random Search) on the continuous benchmark in Figure 5. The cumulative performance on the continuous surrogate tasks matches well with the performance of these methods on the discrete tasks (Figure 2). In particular, we highlight a similar comparative trend of Deep Kernel GP being marginally better than GP after many trials but significantly superior to Random Search.

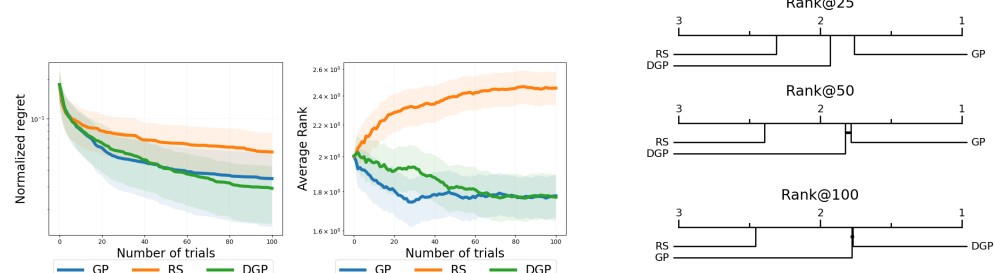

Figure 5: **Aggregated** comparisons of normalized regret and mean ranks across all search spaces for three typical non-transfer HPO methods on the continuous search spaces of HPO-B-v3

## 8 Limitations of HPO-B

A limitation of HPO-B is that it only covers black-box HPO tasks, instead of other HPO problems, such as grey-box/multi-fidelity HPO, online HPO, or pipeline optimization for AutoML libraries. In addition, HPO-B is restricted by the nature of search spaces found in OpenML, which contains evaluations for well-established machine learning algorithms for tabular data, but lacks state-of-the-art deep learning methods, or tasks involving feature-rich data modalities (image, audio, text, etc.). An additional limitation is the structured bias and noise produced by relying on a surrogate for constructing continuous search spaces. However, it has been found that tree-based models are able to model the performance of several machine learning algorithms and produce surrogates that resemble real-world problems [11]. Other sources of bias and noise might come from the user-oriented data generation process for the evaluation on discrete search spaces, which might potentially incur in wrong values or hyperparameters within ranges reflecting prior knowledge or typical human choices. These risks can be reduced by benchmarking on a large number of search spaces, as we suggested throughout the paper.

## 9 Conclusions

Recent HPO and transfer-learning HPO papers inconsistently use different meta-datasets, arbitrary train/validation/test splits, as well as ad-hoc preprocessing, which makes it hard to reproduce the published results. To resolve this bottleneck, we propose HPO-B, a novel benchmark based on the OpenML repository, that contains meta-datasets from 176 search spaces, 196 datasets, and a total of 6.4 million evaluations. For promoting reproducibility at a *level playing field* we also provide initial configuration seeds, as well as predefined training, validation and testing splits. Our benchmark contains pre-processed meta-datasets and a clear set of HPO tasks and exact splits, therefore, it enables future benchmark results to be directly comparable. We believe our benchmark has the potential to become the *de facto* standard for experimentation in the realm of black-box HPO.

## Acknowledgements

The research of Hadi S. Jomaa is co-funded by the industry project "IIP-Ecosphere: Next Level Ecosphere for Intelligent Industrial Production". Prof. Grabocka is also thankful to the Eva Mayr-Stihl Foundation for their generous research grant, as well as to the Ministry of Science, Research and the Arts of the German state of Baden-Württemberg, and to the BrainLinks-BrainTools Excellence Cluster for funding his professorship. In addition, we thank Arlind Kadra for his assistance in interfacing with the OpenML Python package.

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
