# OpenReview forum: "HPO-B: A Large-Scale Reproducible Benchmark for Black-Box HPO based on OpenML "
_NeurIPS.cc/2021/Track/Datasets_and_Benchmarks/Round2 — NeurIPS 2021 Datasets and Benchmarks Track (Round 2)_

### Official Review · Reviewer_5Jpm · 2021-09-20
**HPO-B: A Large-Scale Reproducible Benchmark for Black-Box HPO based on OpenML**

**Rating:** 6
**Confidence:** 3
**Correctness:** Good
**Clarity:** Above the average

**Strengths:**

The authors studied a very urgent problem, especially in the finetuning process of transfer learning. The black-box HPO is a core problem. However, previous methods based on parametric surrogate models for estimating the performance of a configuration overfit the limited number of evaluated configurations. The impact of black-box HPO algorithms is hindered by their poor experimental reproducibility. The proposed HPO-B is a large-scale reproducible benchmark for Black-Box HPO based on OpenML. It seems to be very promising.

The experiments in this paper seem to be strong.


**Weaknesses:**

However, there are some concerns about HPO-B. As mentioned by the authors, it is restricted by the nature of search spaces found in OpenML, which contains evaluations for well-established machine learning algorithms for tabular data, but lacks state-of-the-art deep learning methods, or tasks involving feature-rich data modalities (image, audio, text, etc.).

**Additional Feedback:**

N/A

**Documentation:**

Above the average

**Relation To Prior Work:**

Above the average

**Summary And Contributions:**

In this paper, the authors proposed HPO-B, which seems to be the largest public benchmark of meta-datasets for black-box HPO containing 6.4M hyperparameter evaluations across 176 search spaces (algorithms) and on 196 datasets in total.

The authors studied a very urgent problem, especially in the finetuning process of transfer learning. The black-box HPO is a core problem. However, previous methods based on parametric surrogate models for estimating the performance of a configuration overfit the limited number of evaluated configurations. The impact of black-box HPO algorithms is hindered by their poor experimental reproducibility. The proposed HPO-B is a large-scale reproducible benchmark for Black-Box HPO based on OpenML. It seems to be very promising.

The experiments in this paper seem to be strong.

However, there are some concerns about HPO-B. As mentioned by the authors, it is restricted by the nature of search spaces found in OpenML, which contains evaluations for well-established machine learning algorithms for tabular data, but lacks state-of-the-art deep learning methods, or tasks involving feature-rich data modalities (image, audio, text, etc.).

---

### Official Review · Reviewer_9DY6 · 2021-09-23
**Good HPO benchmark focused on reproducibility**

**Rating:** 8
**Confidence:** 4

**Strengths:**

I think this benchmark, including the provided code and guidelines, are of value to the community.
On the one hand it greatly simplifies evaluation of new HPO methods (in that the evaluation procedure is (almost) completely predefined), on the other it allows for low budget evaluations (with metadataset/surrogate model) and out-of-the-box comparison to earlier results (due to using the same setup).

The use of surrogate models and re-usability of previous evaluations should also lead to fewer wasted resources for new evaluations and comparisons to previous work.


**Weaknesses:**

My main concern is with V3, the transfer-learning evaluation, consisting of only one train/validation/test split.
With only one split and as few as 2-3 datasets in the test set in extreme cases, I feel that results might be influenced by this selection.
It would be good to extend V3 with multiple splits and/or evaluate how sensitive final HPO performances/rankings are to the choice of split.

In some cases the test set, in terms of number of evaluations, is almost as big as the training set (svm (6)) while in others the test set is only roughly 10% (rpart.preproc (16)). It is not clear to me if this is problematic and/or a point of improvement (I didn't have time to mull this over), but perhaps it's worth addressing.

In Section 5.4 you introduce the surrogate models to address the need for (predicted) response values in a continuous search setting.
What remains implicit is that the discrete hyperparameters are evaluated exhaustively, is that correct?
Is there any inspection done as to how many configurations are evaluated for each unique combination of discrete hyperparameter configurations?
Because it seems like the response from different continuous hyperparameters would depend on the setting of the discrete ones.

**Additional Feedback:**

Please excuse my review for being brief, I was called in last minute.
In particular I have also not yet had the opportunity to have a good look at the code/tools provided, or the reviews on the previous version, or even to sit down to mull things over.

**Clarity:**

The paper is well written, and managed to provide answers to most questions that arose while reading it. Though some of it seems needlessly complicated (in Table 2 'benchmark' is redefined, some terms are not used at all (e.g. meta-instance, meta-task)).
L134 was confusing to me because it read as though the splits were defined within the meta-datasets instead of across them, and L165-167 was confusing to me because in the table above only R-based search spaces were shown (but from reading the appendix I gather it's about the V1 set).

**Correctness:**

As far as I can tell it seems appropriate, though it would be better if a motivation was provided for the choice to stop at 100 evaluations.

**Documentation:**

From cursory glance of the repository and descriptions in the paper, yes.

**Ethics:**

No.

**Relation To Prior Work:**

Due to the explicitly mentioned connection with AutoML, it feels like [1] should not be excluded in L#39.
While explicitly not required (because it's recent work, only on ArXiv after the submissions deadline), a comparison to HPOBench ([2]) seems highly relevant.

[1] https://arxiv.org/abs/1907.00909
[2] https://arxiv.org/abs/2109.06716

**Summary And Contributions:**

The authors propose a benchmark for hyperparameter optimization, both in-task (non-transfer) and across tasks (transfer).
To this end they have compiled a set of curated sets of results from OpenML, and created three benchmark sets each with a different goal:
 v1: all of the data, including search spaces and datasets with limited configurations
 v2: a subset of v1 where each configuration is evaluated on at least 10 datasets and each dataset has at least 100 evaluations
 v3: a division of the meta-datasets in v2 to allow for replicable transfer-learning setups

The authors detail the experimental setup and evaluation that should be used when using the benchmark to ensure comparability of results.
Finally they demonstrate the benchmark and its guidelines by conducting evaluations with well known HPO methods.

---

### Official Review · Reviewer_3D2V · 2021-09-24
**New and Large-Scale Benchmark for Non-transfer and Transfer Black-Box HPO**

**Rating:** 8
**Confidence:** 2
**Clarity:** This paper is clearly written and all…

**Strengths:**

The strengths of the HPO-B benchmark (and its variants) include reproducibility, because the authors outline a clear and relatively simple experimental protocol for comparing HPO methods, and scale. The size of this benchmark ensures that many HPO methods for exploring models on low-feature data are included. Since HPO-B is accompanied by different versions that accommodate non-transfer and transfer black-box HPO methods, this benchmark also boasts generalizability and outstanding reach in the HPO method domain. This benchmark is promising in that it may serve as an inspiration for benchmarks for HPO methods concerning models of different data modalities as well. Thus, a major strength of this benchmark is that it demarcates a way of juxtaposing various HPO methods and may very well be the first of its kind.

**Weaknesses:**

The primary weaknesses of this benchmark include underwhelming description of preprocessing HPO evaluations for the benchmark and application under certain modalities. The authors should have described their motivations for certain preprocessing choices and perhaps should have given a more in-depth explanation of the evaluation metrics they recommend when comparing new black-box HPO methods, along with the motivation for these metrics.

In particular, the authors state considerable manual effort was put into deciding which hyperparameters required log-scaling when preprocessing hyperparameter configurations and the associated evaluations for the benchmarks. What entailed significant manual effort? What was the protocol for inspecting the hyperparameter distributions and making the decision to log-scale?

Why was average normalized regret and average rank used as the metric for evaluating HPO methods?

**Additional Feedback:**

N/A

**Correctness:**

This submission contains largely correct claims. Experimental design for testing comparison quality with the recommended experimental protocol seems sound, although motivation for evaluation metrics can be made more clear. Also, it is unclear why 100 evaluations is a generally good place to stop evaluating HPO methods. Is 100 trials the standard for investigating an HPO algorithm's response over different configurations?

**Documentation:**

This benchmark comes with a clear protocol for reporting future comparisons between black-box HPO methods, but the evaluation collection stage for actually procuring the benchmarks could be explained more in-depth. In addition, motivation for preprocessing the evaluations in the stated manner for the benchmark should be made more clear.

**Ethics:**

The authors claim that this benchmark serves as a level playing field for HPO methods, and it seems as though there are no ethical concerns with regards to bias towards specific HPO methods.

**Relation To Prior Work:**

The authors make it a point to emphasize the novelty and potential impact of the HPO-B benchmark. It serves as a large-scale, reproducible platform for comparing black-box HPO methods, which has not been clearly attempted before. Their work clearly differs from previous contributions because it is the first of its kind as it is known.

**Summary And Contributions:**

The authors claim to have assembled the largest benchmark for comparing black-box hyperparameter optimization (HPO) methods or algorithms known as HPO-B. Not only is HPO-B the largest benchmark of its kind, but the authors also defend its reproducibility through exemplifying well-delineated experimental protocols, data splits, and evaluation metrics. In addition to the primary benchmark HPO-B, the authors also contribute two other ready-to-use variants of the benchmark that are better suited for comparing HPO methods under transfer experiments. All three benchmarks boast HPO experiment diversity, and the authors believe these benchmarks will significantly accelerate and catalyze future research in black-box HPO.

The authors first begin with an overview of the greater hyperparamater optimization problem and clearly define the subproblem they address: the function of performance of an HPO algorithm with respect to the hyperparameters themselves is unknown. They describe and cite various existing non-transfer and transfer black-box HPO algorithms and sparsely delineate the issues with directly comparing such methods. The authors continuously underline the pertinence of their benchmark as a remedy to the reproducibility problem that plagues any comparison study of HPO methods.

There is a slight deviation of the work where the authors explain Bayesian optimization concepts as it is one of the more popular black-box HPO methods. The three benchmarks are then clearly described and the purposes of each. After a general summarization of data collection and the datasets, the authors report experimental results for a recommended experimental protocol for standardizing the comparison between HPO methods, a significant contribution.

Limitations of the benchmark are briefly indulged and the work is concluded.

This work has the potential to be a gold-standard for HPO.

---

### Official Review · Reviewer_ajXE · 2021-09-24
**A Comprehensive and Usable HPO Benchmark**

**Rating:** 7
**Confidence:** 4

**Strengths:**

&nbsp;

1. The benchmark is comprehensive and the codebase is well-documented.

2. There are recommended standards for reporting results on the benchmark.

3. There is a usage example provided in the README of the GitHub repo (I have some suggested minor amendments to this below).

&nbsp;

**Weaknesses:**

&nbsp;

## __MAJOR POINTS__

&nbsp;

1. In the Usage section of the GitHub repo it may be beneficial to include an example implementing and testing a custom algorithm e.g. even something as simple as random search. For example, the authors instruct users to: "Create a class that encapsulates the new HPO method.". It might be nice to have an example for this step.

2. In the description of Bayesian optimisation, it may be worth mentioning that the response noise may also be heteroscedastic [1] in many real-world tasks including HPO [2].

&nbsp;

## __MINOR POINTS__

&nbsp;

1. Are [3] the first to propose GP surrogates for HPO? I can't seem to find earlier references so this may indeed be the case! The work in [4] may also be worth mentioning here.

2. For the tables in the main paper it would be great to have the captions included above the table. Apparently the motivation for this is explained here! https://tex.stackexchange.com/questions/3243/why-should-a-table-caption-be-placed-above-the-table

3. The updated version of reference 6 in the submission now has a new title. [2]!

4. Would it be worth renaming the versions of the HPO benchmark in line with the tasks they represent i.e. instead of HPO-B-v2 to call this HPO-B-NT with NT an acronym for Non-Transfer?

5. For the rank plots in Figures 1-5 would it be possible to include more space between the numerical figures to make the plots more readable? Feel free to ignore this comment if it is a lot of work!

6. As typically happens with Bayesian optimisation paper supplements, there appear to be some rendering issues with the supplementary material (caused by the extensive regret plot figure sets)! It might be worth attempting to compress these figures for the camera-ready version.

&nbsp;

## __REFERENCES__

&nbsp;

[1] Griffiths et al. Achieving Robustness to Aleatoric Uncertainty with Heteroscedastic Bayesian Optimisation, Machine Learning: Science and Technology. 2021

[2] Cowen-Rivers et al. An Empirical Study of Assumptions in Bayesian Optimisation. arXiv preprint arXiv:2012.03826. 2020.

[3] Bergstra et al. Algorithms for Hyper-Parameter Optimization. Advances in neural information processing systems. 2011.

[4] Snoek et al. Practical Bayesian Optimization of Machine Learning Algorithms. Advances in neural information processing systems. 2012.

[5] Eggensperger et al. HPOBench: A Collection of Reproducible Multi-Fidelity Benchmark Problems for HPO. arXiv preprint arXiv:2109.06716. 2021.

&nbsp;


**Additional Feedback:**

&nbsp;

All comments included in the main response.

&nbsp;

**Clarity:**

&nbsp;

The paper is well written and easy to follow.

&nbsp;

**Correctness:**

&nbsp;

To the best of my knowledge the results reported by the authors are correct.

&nbsp;

**Documentation:**

&nbsp;

Datasheet provided in the SI. A release and a DOI for the codebase might be beneficial.

&nbsp;

**Ethics:**

&nbsp;

Adequately addressed.

&nbsp;

**Relation To Prior Work:**

&nbsp;

The work of [5] also introduces a HPO benchmark. Understandably the authors would not have been able to include this reference in the submitted version but it may be worth referencing in the camera-ready version!

&nbsp;

**Summary And Contributions:**

&nbsp;

The paper introduces a new Hyper-Parameter Optimisation (HPO) benchmark comprising 176 algorithms and 196 datasets. There is a particular focus on transfer learning HPO.

&nbsp;

---

### Official Review · Reviewer_u9Bi · 2021-09-24
**HPO-B Review**

**Rating:** 6
**Confidence:** 3

**Strengths:**

- **Efficiency**: no models had to be run to create the data set by using OpenML
- **Comprehensive**: large number of search spaces and hyperparameter evaluations
- **Simple application**: in python it should be fairly simple to evaluate new algorithms by simply replacing few lines of code due to the provided examples

**Weaknesses:**

- **Structured noise/bias in OpenML**: I can imagine several problems with the extraction of run results from OpenML. Some that come to my mind in the following points. Could you elaborate on how you overcame these problems?
    - There is significant correlation between search spaces, e.g, your data set contains results of random forests implemented in weka, mlr, and sklearn
    - The runs could have been conducted with different versions of the algorithm, potentially older versions even had bugs.
    - It is likely that the results of methods and their hyperparameters published on OpenML are biased. This is because data scientists would probably not consider non-sensical hyperparameter configurations. An algorithm though would potentially if it could. If this is structucally the case, the methods perform worse in reality than in the benchmark.
- **Extraction code unavailable**: while the data set is publicly available along with examples, the construction is intransparent because the code has not been open-sourced. This leads to centralized updates and curation of the data set of the authors although this is technically not necessary.
- **Extension of the considered hyperparameters not easily possible**: while the usage of OpenML is certainly efficient in the sense that no model had to be run to create this benchmark, this is also a major weakness since the hyperparameter configurations are fixed (unless one runs certain models on data sets oneself which is also discouraged).
- **Problem with continuous HPO?** From what I understand, the continuous version of the proposed benchmark is enabled by *learning* an xgboost surrogate. Is this correct? If yes, why do we assume that this models the response well enough? In my opinion, this will introduce a structural bias due to the choice of xgboost.
- **Unclear naming of data sets and questionable fixed choices**: it is unclear to me why the data sets are called `v1-v3`. Maybe give them a clear name? They all are based on the same data and the choices seem rather arbitrary. In particular, I don't see why it should be discouraged to apply transfer-based approaches to other datasets than in the subset of `v2`. I also can imagine that researchers would be interested in modifying the train/val/test split on `v3`. Although standard benchmarks like ImageNet or CIFAR-X in fact prescribed these splits, I am not sure if this is really a good thing as it might have led to overfitting the validation set over the years.

**Additional Feedback:**

See detailed comments in `Weaknesses`.

**Clarity:**

The paper is mostly well-written and also tries to be accessible to researchers not working on HPO.
One phrasing I stumbled upon is *most frequent search spaces* in line 125: do you mean the meta-datasets with the evaluation of the most hyperparameters? I am not sure if "frequent" is the right word in this case.


**Correctness:**

The claims appear to be correct. The data set is created in a sound way but could have potential flaws/biases for its purpose as pointed out above in `Weaknesses`. The experimental protocol seems reasonable.

**Documentation:**

- The documentation in the form of a python loader exists but it would make sense to describe the json file more precisely so that the use is not limited only to python.
- In my opinion, there is not enough detail on data collection since the code has not been open-sourced and the description of the data cleaning and selection is vague.
- The maintenance plan is unclear. Potentially it would be possible to update the data set regularly but there does not seem to be a plan for this yet.

**Ethics:**

I don't see any immediate ethical concerns.

**Relation To Prior Work:**

Yes.

**Summary And Contributions:**

### Summary
The paper introduces the HPO-B benchmark for black-box hyperparameter optimization (HPO) of machine learning algorithms by drawing from publicly available OpenML runs. Especially but not only, the new benchmark aims to make transfer-learning for AutoML more reproducible by unifying the evaluation.

### Contributions
- Extraction and assembly of results from OpenML into a HPO data set
- Proposed evaluation protocol for future research on black-box and transfer-HPO
- Evaluation of baselines on the proposed data set

### Review summary
Overall, I believe that an HPO benchmark is very valuable and necessary. The proposed methodology to extract and provide data based on OpenML is environmentally friendly, scalable, and efficient. However, this choice comes with key limitations. In my opinion, using only HPO-B in the future for benchmarks could introduce significant evaluation biases due to the data collection biases. For more details, see section `Weaknesses`. I would be happy to discuss this and raise my score if my concerns turn out to be invalid or more minor.

---

### Author Response · Authors · 2021-09-30
**Updated submission**

Dear reviewers,

thank you for your valuable feedback. Following your suggestions, we updated the manuscript with these modifications:

- New section in the Appendix explaining the data-preprocessing in more detail
- New references suggested during the discussions
- Extended discussion on limitations by mentioning other factors such as structured noise and bias
- Fixed other minor points and typos

Moreover, we pushed new changes to our repository:

- Example on how to add a new HPO algorithm
- Script for data extraction and splitting
- Example for pre-processing a new search space
- Explanation of the data format

---

> ### Comment · Reviewer_ajXE · 2021-10-01
> **Many Thanks to the Authors for the Revsisions**
>
> &nbsp;
>
> Many thanks to the authors for the revisions!
>
> &nbsp;

---

### Decision · Program_Chairs · 2021-10-09

**Decision:**

Accept

**Comment:**

All reviewers agree on acceptance. I recommend the authors to look at the comments by the reviewers and use these to improve the paper for the camera-ready version.